# Creating unimolecular multivalent diversity in protein conjugates via the Passerini multicomponent bioconjugation with isocyanoproteins

Ana R. Humpierre[1,2], Yanira Méndez[1,3], Ahyoung Kim[1], Michael Niemeyer[1], Andrej Frolov[1], Mirelys Saenz [2], Raine Garrido [4], Leslie Reguera[1,2], Darielys Santana-Mederos[4], Dagmar Garcia-Rivera[4], Bernhard Westermann [1] ✉ & Daniel G. Rivera [1,2,4] ✉

The ability to conjugate multiple molecules to a protein is of great interest for pharmaceutical and vaccine development, especially if this can be achieved in a one-pot procedure. Multicomponent reactions are powerful procedures that allow the assembly of complex constructs incorporating at least three molecular fragments, but many of them use amino and carboxylic groups that are too abundant in proteins. Herein, we introduce the use of the Passerini 3-component reaction with isocyanoproteins for the assembly of multivalent protein (glyco)conjugates. Proteins were tagged with isocyanide handles and next derivatized to investigate the efficacy and limitations of the Passerini bioconjugation. The multicomponent conjugation enabled the simultaneous functionalization of proteins with two biologically relevant molecules such as carbohydrate antigens, lipids, and polymers. The efficient display of various antigens in a unimolecular multivalent construct is a notable result that paves the way towards new applications in preventive vaccines and therapeutics.

An increasing number of pharmaceuticals rely on the conjugation of proteins to other molecules, which can be another biomolecule, a cytotoxic drug, a polymer or a fluorescent label, among others[1,2]. Eventually, the conjugation can be done in a site-selective manner to form homogeneous conjugates (Fig. 1A)[3–5], which are typically preferred in the development of therapeutics such as PEGylated proteins, conjugated antibody fragments and antibody-drug conjugates. However, other protein-based pharmaceuticals, such as antibacterial and antiviral conjugate vaccines, as well as nanoparticle vaccines, strongly rely on the assembly of multimeric conjugates to produce a potent immunogenicity[6–11]. In this type of construct, several copies of a hapten or biomolecular antigen, such as a peptide, protein or glycan, are conjugated to a carrier protein for enhancing the immunogenicity of the displayed molecule. Depending on the functionalization level of the antigen, this type of multiple conjugation approach can render either radial conjugates[12,13] (Fig. 1B) or cross-linked macromolecular lattices[8–11]. While both types of constructs are mostly heterogeneous, they serve the purpose of eliciting a potent immune response against the antigen attached to the carrier protein.

In immunology, multivalency is crucial to understanding how nature boosts the activation of immune cells through the multivalent display of antigens, which strengthens binding compared to a monovalent receptor-ligand interaction[14]. The term multivalency is also used in antibacterial glycoconjugate vaccines, such as pneumococcal and meningococcal vaccines, for formulations incorporating several protein-polysaccharide conjugates, each one containing a glycan of a different bacterial serotype (or serogroup)[8–13]. In this context, the vaccine formulation is multivalent (the term 'polyvalent' is also used indistinctly), but each glycoconjugate is monovalent concerning the bacterial polysaccharide against which the immune response is elicited. It is known that carbohydrates are T-cell independent antigens and they need to be covalently linked to a carrier protein or an immune-stimulating lipid for eliciting a long-lasting and protective immunity[8–11,15,16]. Although the multivalent display of glycans—and to a lesser extent peptides and proteins—attached either to a carrier or nanoparticle protein is key for a potent immunogenicity, such multimeric conjugates do not possess multivalent diversity as they typically display multiple copies of the same antigen[8–18]. Strictly speaking, the attachment of

[1]Department of Bioorganic Chemistry, Leibniz Institute of Plant Biochemistry, Weinberg 3, Halle/Saale, Germany. [2]Laboratory of Synthetic and Biomolecular Chemistry, Faculty of Chemistry, University of Havana, Zapata & G, Havana, Cuba. [3]Yusuf Hamied Department of Chemistry, University of Cambridge, Cambridge, United Kingdom. [4]Finlay Institute of Vaccines, Ave 27 Nr. 19805, Havana, Cuba. ✉e-mail: Bernhard.Westermann@ipb-halle.de; dgr@fq.uh.cu

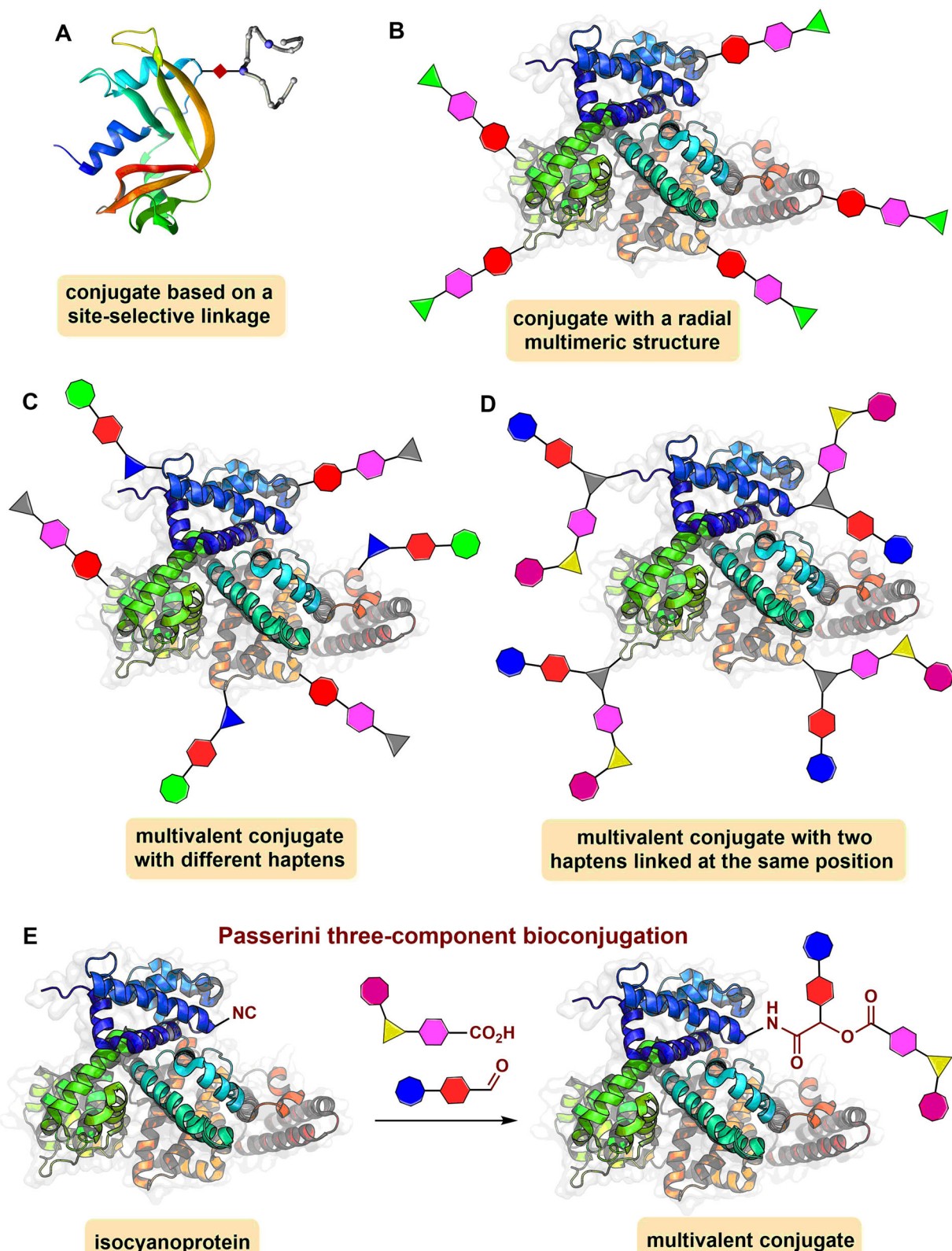

**Fig. 1 | Representative examples of protein conjugates. A** Conjugate derived from a single site-selective ligation of a molecule/hapten to a protein. **B** Multimeric conjugate bearing multiple copies of the same molecule/hapten (radial type). **C** Multivalent conjugates bearing two dissimilar molecules/haptens attached at different positions via multistep protocols. **D** Multivalent conjugate bearing two different molecules/haptens attached at the same position. **E** This work: Passerini three-component bioconjugation leading to multivalent conjugates.

at least two different antigenic components—even if they are of the same biomolecular nature—to the carrier protein or nanoparticle is required for generating multivalent diversity in a bioconjugate (Fig. 1C).

In vaccinology, there is a great interest in developing strategies toward protein conjugates or nanoparticles capable of displaying different types of antigens to the immune system (Fig. 1C, D). A notable example is the use of the SpyTag/SpyCatcher[17] conjugation technology to produce multivalent coronavirus vaccines presenting multiple viral protein antigens attached to a carrier protein nanoparticle[19,20]. This strategy has led to potent antibody responses against all protein antigens displayed in the nanoparticle conjugates. Antibacterial multivalent glycoconjugates bearing either two different bacterial polysaccharides[21] or a polysaccharide antigen and a toll-like receptor (TRL) agonist[22] attached to the carrier protein have also been proposed as vaccine candidates (Fig. 1C). The production of bioconjugates displaying a hapten/antigen and an adjuvant molecule at the same time is also of great interest for the development of self-adjuvanting anticancer vaccines. For example, highly immunogenic conjugates have been built by attaching a TRL ligand and a tumor-associated glycopeptide antigen to the bovine serum albumin (BSA) protein[23,24]. These examples and others not based on carrier proteins[25] are often referred to as unimolecular multicomponent vaccines, as they incorporate three different components. However, the synthetic and bioconjugation strategies used to produce them are not multicomponent, as they require laborious, stepwise conjugation protocols with various purification steps.

We have contributed to the field of unimolecular multivalent vaccines with the development of a Ugi multicomponent glycoconjugation strategy capable of conjugating two different oligo/polysaccharide antigens to a carrier protein in a one-pot process (Fig. 1D)[26,27]. Using this technology based on the Ugi four-component reaction (U-4CR), we have produced bivalent glycoconjugate vaccines incorporating bacterial capsular polysaccharides either from two different pneumococcal serotypes or from different bacteria (Gram-positive and Gram-negative) into the tetanus toxoid carrier protein[28,29]. Our group[26–32] and others[33–37] have shown a variety of applications of isocyanide-based multicomponent reactions (MCRs) in the conjugation of peptides and proteins, but with the biomolecule reacting always either by their canonical amino or carboxylic groups or by previously introduced hydrazide or oxo-functionalities.

Here we describe a new type of multicomponent bioconjugation by introducing the Passerini three-component reaction (P-3CR) with isocyanoproteins as a versatile strategy to functionalize proteins in a multivalent manner (Fig. 1E)[38,39]. Unlike the U-4CR, which requires the preformation of an imine intermediate that can be challenging in water, the P-3CR leads to a multivalent conjugate by the one-pot reaction of an isocyanide-tagged protein with other molecules functionalized with an aldehyde or ketone and a carboxylic acid. Although this reaction works best in non-polar solvents like ethers and dichloromethane, various groups have found that it has markedly fast kinetics in water[40–42]. In line with this, we sought to expand the toolbox of isocyanoproteins by showing applications that can benefit from the rapid, multicomponent generation of multivalent diversity. We demonstrate the potential of this strategy by producing novel site-selectively and polyfunctionalized isocyanoproteins, followed by their multicomponent derivatization with biologically relevant molecules such as lipids, polyethylene glycol (PEG), and fluorescent labels, as well as tumor-associated and bacterial carbohydrate antigens typically used in glycoconjugate vaccines.

## Results and discussion

Besides the reports on the Ugi bioconjugation[26–36], other MCRs have been employed to derivatize proteins[43–49], which mostly relied on the modification of highly nucleophilic residues such as Lys and Tyr. However, none of these latter approaches has been used to incorporate two medium-sized or large (bio)molecular components into a protein at once, which is the actual power of a multicomponent protocol. Most of the previous approaches are based on intermediate condensation steps, in which one of the components is typically very small and does not actually contribute to the multivalency of

the conjugate. In this sense, one could argue that using a multicomponent approach to conjugate only one 'relevant' component (e.g., label, probe, hapten, biomolecular antigen, etc.) does not bring any additional value compared to many other conjugation methods that are often very efficient and chemoselective. Based on this rationale, we sought to combine the advantages of introducing a bioorthogonal handle like the isocyanide into proteins with those related to the creation of multivalent diversity in a single multicomponent step.

The interest in isocyanoproteins has recently risen owing to the capacity of this functional group to undergo bioorthogonal reactions in aqueous conditions[50–54]. Leeper and co-workers paved the way for the chemoselective introduction of isocyanide handles into proteins, in this case using the thiol-maleimide chemistry[51]. In addition, a genetic approach has also been described for the site-selective protein functionalization with isocyanides[55]. The isocyanide stability is very high in water at neutral pH, and this can be further improved by using α,α-disubstituted isocyanides, ensuring that hydrolysis to the corresponding formamide is minimized[50]. The bioorthogonality of this functional group relies on its lack of reactivity— per se—with most functional groups present in biomolecules, such as alcohols, amines, thiols, phenols and carboxylic acids. On the other hand, the reaction of isocyanides with carboxylic acids in the presence of either carbonyl (P-3CR) or imine components (U-4CR) is the most powerful feature of this functionality[56,57]. In spite of this, the isocyanide handle has been mostly employed for biomolecule ligation and labeling using cycloaddition reactions[50–54], while to the best of our knowledge, the Passerini reaction has never been explored with isocyanoproteins.

## Preparation of isocyanide-tagged proteins

Scheme 1 depicts our strategy for the site-selective functionalization of proteins with the isocyanide handle, which provides new methods compared to what is reported in the literature. A variety of heterobifunctional isocyanide linkers were synthesized by dehydration of the corresponding formamides, and in all cases, bearing a bioorthogonal reactive group suitable for protein derivatization (Supplementary Methods). Initially, we chose BSA as a model protein for this study, as this carrier protein has proven useful for conjugation to a variety of antigens for the development of conjugate vaccines[23,24,58,59]. With a molecular mass of 66.5 kDa, this immunogenic protein contains 58 Lys—out of which around 10 are surface exposed—16 disulfide bonds, and an unpaired Cys at position 34 that is well suited for conjugation using thiol-maleimide chemistry[60].

Firstly, we explored the incorporation of an isocyanide handle at the protein N-terminus, something that has not been achieved previously. To this end, we employed a method developed by Francis and co-workers[61], which relies on the formation of a functionalized N-terminal imidazolidinone moiety by reaction of 2-pyridinecarboxaldehyde (2PCA) with proteins not bearing a Pro at the second position. The protocol comprises the attack of the neighboring backbone NH to the preformed imine intermediate, thus ensuring the selectivity for the protein N-terminus. Thus, isocyano-2PCAs **1a** and **b** were produced and employed to functionalize with an isocyanide handle various proteins at the N-terminus, including BSA, bovine pancreatic ribonuclease A (RNAse A), and bovine ubiquitin. RNase A is an enzyme that catalyzes the degradation of RNA and is generally employed in plasmid and genomic DNA purification protocols to eliminate carryover RNA. It has a molecular mass of 13.7 kDa and its native structure contains four disulfide bonds, no unpaired Cys and no Pro at position 2. Ubiquitin is a highly preserved protein that occurs in eukaryotic cells, given that ubiquitylation regulates a wide variety of cellular processes, including infection and immunity. Ubiquitin has a molecular mass of 8.6 kDa, and its native structure possesses the same requirements necessary for Francis's approach. The N-terminal derivatization of these proteins was performed at 10 mg mL$^{-1}$ of protein in carbonate buffer at pH 10.4 by incubation overnight at room temperature with 50 equiv of the isocyano-2PCAs **1a** or **1b**. BSA was also site-selectively functionalized by reaction with maleimido-PEG-isocyanide **4** following a standard thiol-maleimide protocol. When this reaction is performed in neutral pH at 4 °C (i.e., PBS at pH 7.4), BSA is

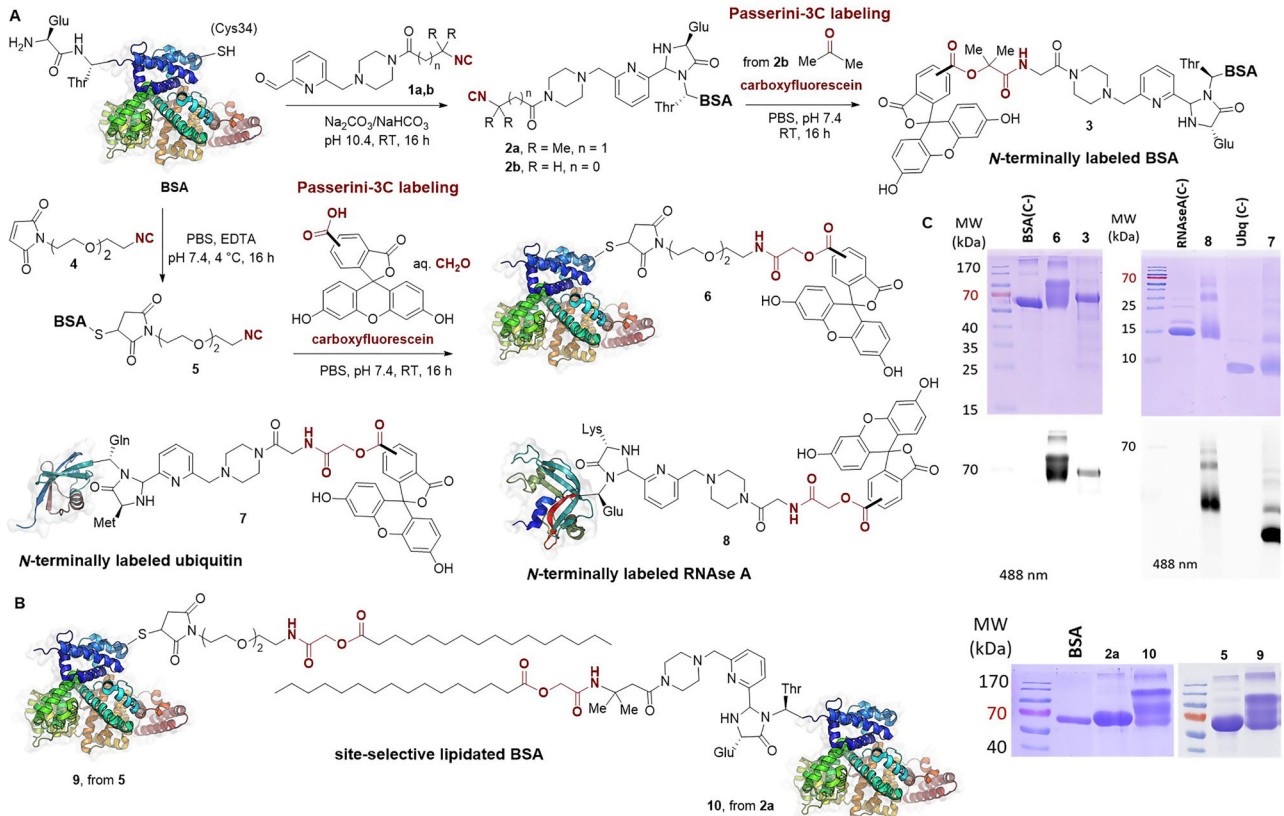

**Scheme 1 | Development of the Passerini-3C bioconjugation with iso-cyanoproteins. A** Site-selective functionalization of the proteins BSA, ubiquitin and RNAse with an isocyanide group and subsequent fluorescent labeling and **B** lipidation. **C** Characterization of the isocyanide-tagged, fluorescently labeled and lipidated proteins by SDS-PAGE with Coomassie staining and in-gel fluorescence scanning in comparison with the native proteins and negative controls BSA(C-),

RNAse A(C-) and ubiquitin (C-) prepared by incubating the native proteins with carboxyfluorescein under the same reaction conditions. Carboxyfluorescein is represented with an ambiguous substitution due to the mixture of regioisomers, and also in the closed spirolactone form, but it should be considered that it also occurs in the open, fluorescent quinoid form.

known to react exclusively at the sulfhydryl group of Cys34[31,62,63]. Protein purification was accomplished by ultrafiltration in Amicon® with 10 kDa cellulose membranes for BSA and in 3 kDa Centricon® for ubiquitin and RNAse A. The incorporation of one isocyanide linker was proven by ESI-TOF or MALDI-TOF mass spectrometry (Fig. S19). Analysis by SDS-PAGE of native BSA in comparison with isocyanide-tagged proteins **2a** and **5** showed that the functionalization reactions take place without protein decomposition, aggregation or cross-linking (Scheme 1C, lower panel).

### Site-selective and multimeric conjugation and labeling

To assess the reactivity of the isocyanoproteins in the Passerini bioconjugation approach, we carried out the multicomponent labeling with carboxyfluorescein of the N-terminal isocyanide-tagged BSA, RNAse A and ubiquitin, as well as the isocyano-BSA **5** functionalized at Cys34. The Passerini reactions were conducted using 50 equiv of carboxyfluorescein and either formaldehyde or acetone as the carbonyl component, at a protein concentration of 2–3 mg mL$^{-1}$ in PBS at pH 7.4 (Scheme 1). The reaction mixtures were conducted protected from light, for 16 h at room temperature, and were terminated using ultrafiltration to remove the low-molecular-weight components. This initial labeling study was relevant to assess the scope and limitations of the Passerini bioconjugation with a variety of iso-cyanoproteins. Analysis by SDS-PAGE with in-gel fluorescence scanning proved the success of the multicomponent labeling by showing fluorescent bands for the four labeled proteins **3**, **6**, **7** and **8** (Scheme 1C, upper panel). Parallel experiments conducted by incubating the native proteins with carboxyfluorescein under the same conditions—with and without formaldehyde —did not lead to any fluorescently labeled conjugate, proving that the reaction only takes place due to the isocyanide tag incorporated. Formation of the

P-3CR-derived conjugates was also confirmed by MALDI-TOF (Fig. S20), which proved the incorporation of the fluorophore, providing a carboxylic acid component. Interestingly, the homogeneity of the labeled protein was not equally satisfactory, as detected in the electrophoresis analysis, which depends on the oxo-component used. For example, isocyanide-tagged BSA **2b**—functionalized at the N-terminus—reacted with acetone and carboxy-fluorescein to produce a labeled protein without noticeable cross-linking. However, a variable amount of protein cross-links was detected when formaldehyde was used as the aldehyde component. As observed in both the Coomassie-stained and fluorescence scanned gels (Scheme 1C), fluorescently labeled BSA **6** was obtained with a significant level of cross-linking after the P-3CR with formaldehyde, despite the fact that the site-selectively tagged isocyanoprotein was quite homogeneous. As a result, we sought to investigate whether the Passerini labeling of smaller isocyanoproteins also comprises certain cross-linking when formaldehyde is employed. To this end, labeled ubiquitin **7** and RNAse **8** were produced from the corresponding N-terminus isocyanide-tagged proteins, also showing a certain level of cross-link formation due to the use of formaldehyde. Previous reports have described that formaldehyde certainly reacts with various nucleophilic amino acid side chains, leading to protein cross-links[64], a reactivity pattern that has been widely exploited in the detoxification process for converting tetanus and diphtheria toxins into toxoids[65,66].

We also implemented a Passerini lipidation protocol by reacting isocyanide-tagged BSA **2a** and **5** with palmitic acid and formaldehyde, leading to the palmitoylated proteins **9** and **10**. The unequivocal incorporation of the lipid tails was confirmed by MALDI-TOF. However, the optimization of the lipidation procedure was more challenging than the fluorescent labeling due to the poor solubility of palmitic acid; so only 20

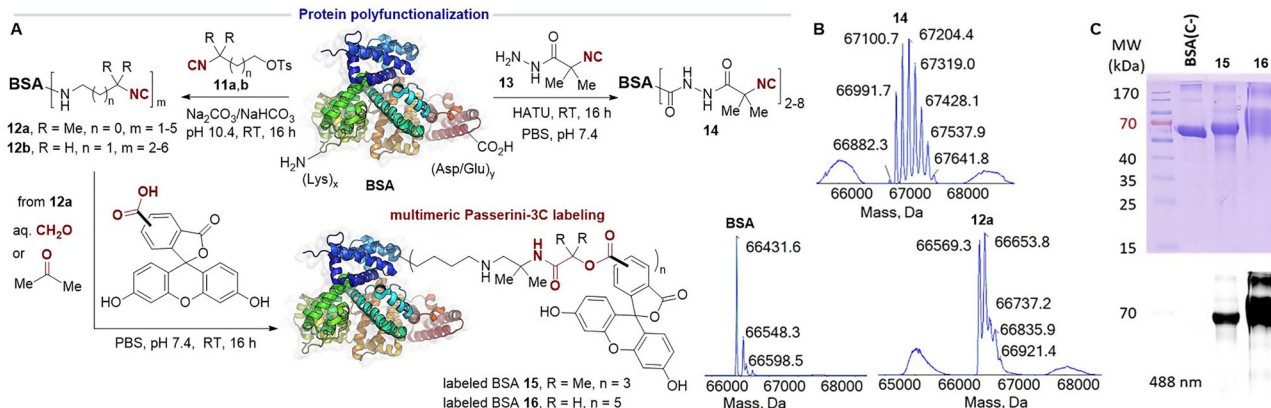

**Scheme 2 | Protein polyfunctionalization and labeling by the P-3C bioconjugation. A** Derivatization of BSA with multiple isocyanide groups and subsequent multicomponent fluorescent labeling. **B** ESI-TOF mass spectra of native BSA and isocyano-BSAs **12a** and **14**. **C** Characterization of the fluorescently labeled BSA **15** and **16** by SDS-PAGE with Coomassie staining and in-gel fluorescence scanning in comparison with a negative control BSA(C-) prepared by incubating BSA with carboxyfluorescein under the same reaction conditions.

equiv of the fatty acid was used to avoid turbidity in the reaction. Protein aggregates were detected in the electrophoresis analysis of the two BSA lipoconjugates. In this specific case, this could also be due to the great lipid-binding capacity of BSA, which can certainly prompt further protein aggregation.

To produce multimeric protein conjugates instead of site-selectively modified ones, protocols enabling the polyfunctionalization of proteins with isocyanide groups were implemented. We prepared the heterobifunctional linkers **11a** and **11b** bearing an alkyl isocyanide functionalized with a tosyl group to enable the alkylation of BSA at the amino groups of Lys and the N-terminus (see the SI). As shown in Scheme 2A, the reactions were carried out at 10 mg mL$^{-1}$ of protein concentration in carbonate buffer at pH 10.4, overnight and with an excess of the tosylated heterobifunctional linkers. Analysis by ESI-TOF mass spectrometry (Scheme 2B) showed the incorporation of various alkyl isocyanide moieties into the protein. Isocyanide-tagged BSA **12a** incorporated up to 5 units of the alkyl isocyanide, with the protein bearing 1 and 2 units as the major products, as suggested by ESI-MS (Scheme 2B). The use of the less sterically hindered isocyano-alkyl tosylate **11b** led to the incorporation of up to 6 isocyanide groups into BSA **12b**, with the major products incorporating 2, 3 and 4 isocyano-alkyl units. We also sought to exploit the reactivity of carboxylic acids for the polyfunctionalization of BSA with isocyanide groups. To this end, we synthesized the hydrazido-isocyanide **13** from the commercially available α-aminoisobutyric acid methyl ester by formylation, dehydration and hydrazide formation (see the SI). Activation of BSA carboxylic acids with hexafluorophosphate azabenzotriazole tetramethyl uronium (HATU) and treatment with **13** enabled the incorporation of up to 8 isocyanide groups into the protein **14**, with product bearing 2, 3, 4, 5 and 6 hydrazido-isocyanide units as major products, as suggested by ESI-MS (Scheme 2B).

For comparison purposes, we explored the reactivity of the polyfunctionalized BSA **12a** in the Passerini labeling, using the same conditions as before but with 50 equiv of carboxyfluorescein and either formaldehyde or acetone as oxo components. The successful formation of the multimeric, fluorescently labeled BSA **15** and **16** was confirmed by MALDI-TOF, which also allowed estimating the amount of carboxyfluorescein units incorporated into the protein. The P-3CR between carboxyfluorescein and BSA, using acetone incorporated up to 3 units of the fluorophore (BSA **15**), while using formaldehyde allowed introducing up to 5 units (BSA **16**). While the bioconjugation reaction with acetone led to a lower functionalization, it was certainly better than that of formaldehyde in terms of homogeneity. As shown in Scheme 2C, analysis by SDS-PAGE of the labeled proteins **15** and **16**, in comparison with BSA incubated only with carboxyfluorescein, once again demonstrated that formaldehyde—but not acetone—provokes a notable protein cross-linking.

Although the participation of a neighboring carboxylic acid (i.e., Glu or Asp residue close to the isocyanide position) in an intramolecular P-3CR cannot be fully discarded, our findings confirmed the unequivocal incorporation of the molecule providing the carboxylic acid. The conditions optimized for the P-3CR include an excess of the external carboxylic acid and the oxo-component, which, together with the relatively low protein concentration, seems to be sufficient also to rule out any possibility of intermolecular P-3CR with another protein molecule. Overall, our initial experiments with site-selectively and polyfunctionalized isocyanoproteins proved that the Passerini bioconjugation is a suitable method for protein derivatization and labeling, although the protocol is not recommended with formaldehyde as a carbonyl component. Acetone or any other reactive aldehyde is preferable for producing Passerini-modified proteins, especially if the aldehyde introduces another diversity element, as the carboxylic acid component does.

## One-pot assembly of multivalent conjugates

In the pursuit of medicinal applications, we turned to exploit to the maximum the multicomponent nature of the Passerini bioconjugation by using both carbonyl and carboxylic acid components of biological relevance. Certainly, there is a high availability of immunologically and pharmacologically important molecules functionalized with these groups, which could be incorporated into proteins in one shot. Scheme 3 depicts the strategy using the P-3CR for conjugating a protein simultaneously to a carbohydrate antigen and a lipid or PEG, at positions that were previously tagged with an isocyanide handle.

Initially, we carried out the conjugation of isocyano-tagged BSA to sialic acid (i.e., N-acetylneuraminic acid) along with either a lipid or a 2 kDa PEG functionalized with an aldehyde group. Sialic acid is a carbohydrate unit typically found on cell surfaces at the end of glycan chains—e.g., in glycoproteins and glycolipids—playing a crucial role in signaling, cell and pathogen recognition, immune regulation, and brain development. Accordingly, we turned to producing glycoproteins in which the sugar unit is linked to the protein together with another molecule of immunological (lipid) or pharmacological interest (PEG). The P-3CRs were carried out at 3 mg mL$^{-1}$ of protein in PBS at pH 7.4, room temperature for 24 hours and with a 50-fold excess of sialic acid and aldehyde component. The resulting conjugates were purified by ultrafiltration with 10 kDa cellulose membranes. Thus, the site-selectively tagged isocyano-BSA **5** and **2b** were sialylated and lipidated at the same time, producing the lipidic glycoconjugates **17** and **18**, respectively (Scheme 3B). In parallel, isocyano-BSA **5** was simultaneously PEGylated and glycated with sialic acid to produce the site-selective conjugate **19**, while isocyano-BSA **12b** reacted with the same components to produce conjugate **20** featuring a multimeric lipo-glyco-protein structure. Electrophoresis

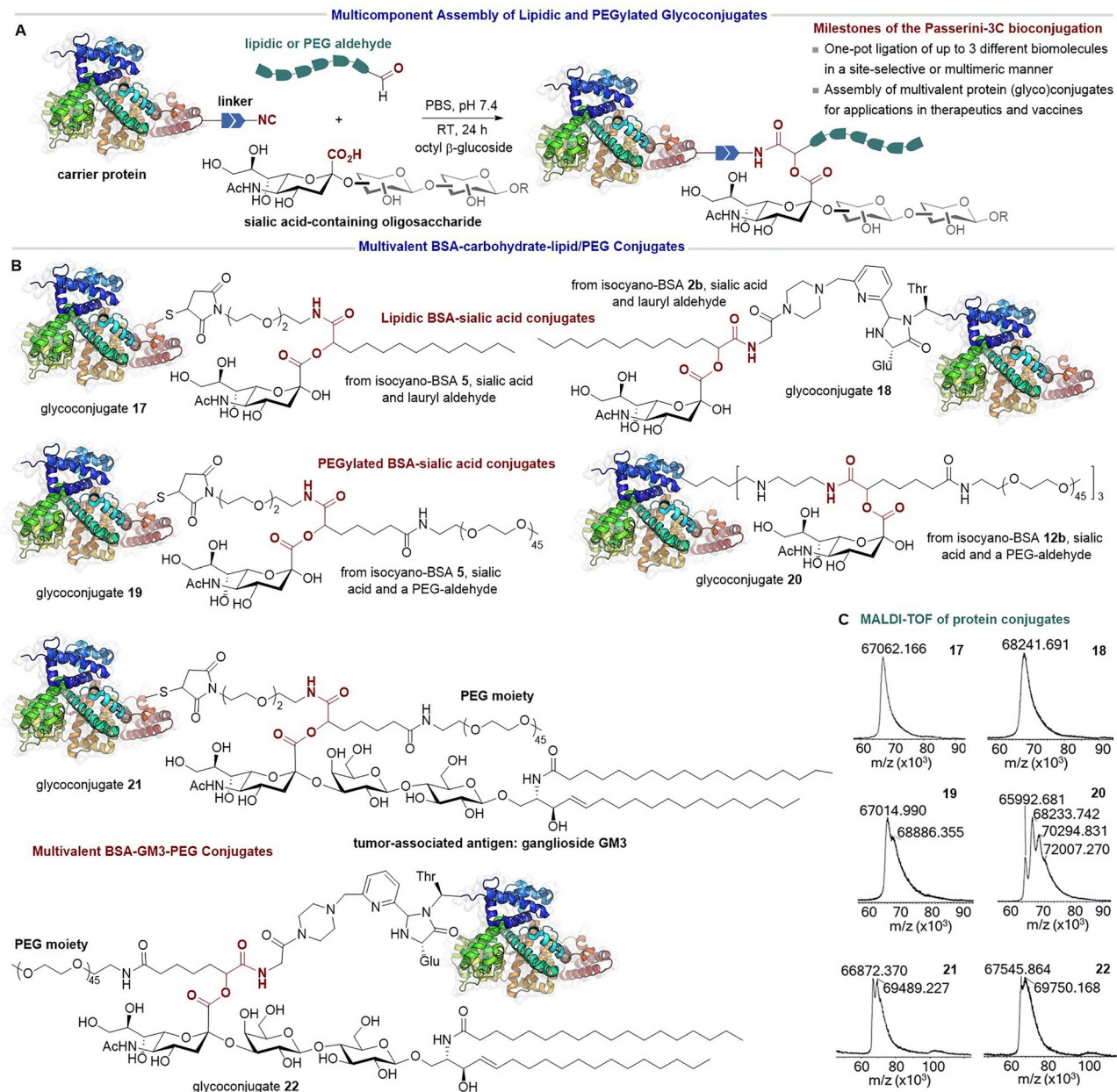

**Scheme 3 | Passerini bioconjugation for the assembly of multivalent protein conjugates**. **A** Multicomponent conjugation of a carbohydrate antigen and either lipidic or PEGyl moiety to a carrier protein. **B** Structure of the multivalent protein conjugates incorporating BSA, sialic acid, the tumor-associated ganglioside GM3, lipids and PEGs. **C** MALDI-TOF mass spectra of the lipidic and PEGylated glycoconjugates.

analysis proved that no protein cross-linking or degradation takes place in the presence of the lauryl or PEG aldehyde (Fig. S21). In addition, we were positively impressed by the good reactivity of sialic acid in this multi-component bioconjugation protocol, considering the rather sterically hindered nature of its carboxylate group. The MALDI-TOF mass spectrometry analysis (Scheme 3C) showed that glycoconjugates **17**, **18** and **19** incorporated one Passerini adduct, while conjugate **20** added up to three units of Passerini products, as it arises from a polyfunctionalized isocyanoprotein. An in-depth analysis of the mass spectra indicated that the conjugations with lauryl aldehyde were more efficient than those with the PEG aldehyde, as no evidence of unreacted isocyanoproteins was detected in the protocols using the former component. This is reasonable due to the lower reactivity of the aldehyde bearing a very long PEG chain.

To further illustrate the potential of the multicomponent assembly strategy, we focused on the conjugation of the carrier protein to a complex sialic acid-containing oligosaccharide and PEG fragment capable of increasing metabolic stability. This was achieved using the tumor-associated carbohydrate antigen GM3[67], a glycolipid produced as previously reported by a team from the Finlay Institute of Vaccines[68]. GM3 is a ganglioside composed of a ceramide backbone (i.e., a sphingosine base N-acylated with a fatty acid) and a trisaccharide fragment made up of β-D-glucose, β-D-galactose and terminal sialic acid (Scheme 3B). This glycosphingolipid is a relevant therapeutic target owing to its properties as a regulator in cancer progression, metabolic diseases, immune responses, and neurological development. As shown in Scheme 3B, we produced multivalent conjugates **21** and **22** featuring a level of structural complexity that is difficult to achieve by any other bioconjugation method in only one step. These conjugates incorporate BSA site-selectively modified at either the N-terminus or the residue Cys34, together with the complex glycolipid skeleton of GM3 and a long PEG chain. However, because of the complex amphiphilic structure of the glycolipid, the conjugation protocol required further optimization of the conditions used before, consisting of the use of octyl-β-glucoside as a

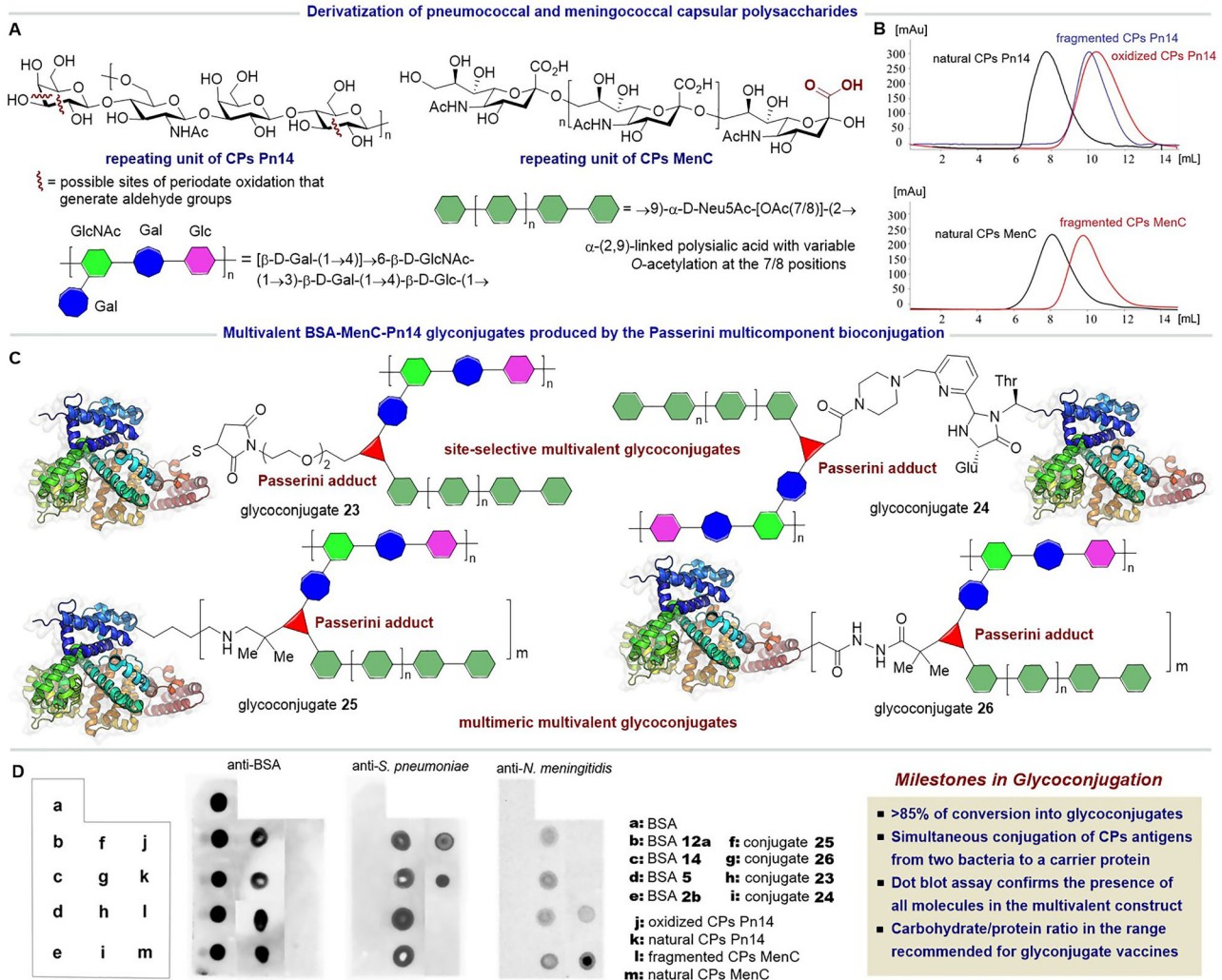

**Scheme 4 | Passerini bioconjugation for the assembly of multivalent glyco-conjugates incorporating bacterial polysaccharide antigens.** A Repeating unit of the pneumococcal serotype 14 (Pn14) and the meningococcal serogroup C (MenC) capsular polysaccharides (CPs). B SE-HPLC traces (TSK 5000 PW column) of the natural, fragmented and—for CPs Pn14—oxidized polysaccharides. C Schematic representation of the structures of the multivalent BSA-MenC-Pn14 glycoconjugates produced by the Passerini bioconjugation with isocyanoproteins. D Antigenicity evaluation by Dot blot assays of glycoconjugates 23, 24, 25 and 26 in comparison with isocyano-BSAs and modified CPs, using reference antibodies against BSA, *S. pneumoniae* and *N. meningitidis*. Native BSA and the natural CPs antigens were used as positive controls.

detergent to solubilize the GM3 and the reduction of the protein concentration to 1.5 mg mL$^{-1}$. The resulting conjugates were purified by dialysis following an ultrafiltration step. The MALDI-TOF spectra of conjugates 21 and 22 are consistent with the incorporation of one Passerini adduct, including GM3 and the PEG (Scheme 3C), while no protein cross-linking was observed in the SDS-PAGE (Fig. S21).

## Multivalent glycoconjugates incorporating two bacterial polysaccharide antigens

The development of commercial antibacterial glycoconjugate vaccines has mostly relied on three types of conjugation methods: i) cyanylation, ii) reductive amination and iii) thiol-maleimide chemistry[8–13,69,70]. All these strategies allow the incorporation of one type of poly/oligosaccharide chain per conjugation step, so multiple procedures are required to achieve a multivalent formulation. Herein, we sought to evaluate the potential of the Passerini bioconjugation strategy for the preparation of unimolecular multivalent glycoconjugates incorporating polysaccharide antigens of two different bacteria, which is of great interest for the development of combined multivalent vaccines. As shown in Scheme 4A, we chose the capsular polysaccharide (CPs) of *S. pneumoniae* serotype 14 (Pn14)—which is

present in all pneumococcal conjugate vaccines—to be employed as an oxo-component. To this end, CP Pn14 was fragmented to a molecular size of 10–30 kDa and further subjected to periodate oxidation, resulting in approximately one carbonyl group every 10 repetitive units. As shown in Scheme 4A, the repeating unit of CPs Pn14 is a branched tetrasaccharide containing two galactoses (Gal), one glucose (Glc) and one *N*-acetylglucosamine (GlcNAc)[71]. The periodate oxidization of this tetrasaccharide fragments generates aldehyde groups by breaking vicinal diol positions of the galactose and glucose units. Analysis by size exclusion HPLC (SE-HPLC) showed that the fragmented and oxidized CPs14 have similar chromatographic profiles (Scheme 4B), proving that the periodate oxidation did not further modify the size of the polysaccharide. In parallel, the CPs from the serogroup C of *N. meningitidis* (CPs MenC) was fragmented under mild acidic conditions, resulting in a 10–30 kDa polysaccharide antigen already containing the required carboxylic acids (Scheme 4).

Although BSA is not typically used in antibacterial glycoconjugate vaccines, our goal was to assess the efficiency of the P-3C to incorporate the two large CPs into this archetypal carrier protein. As a result, both site-selectively and polyfunctionalized isocyano-BSAs 2b, 5, 12a, and 14 were used in the multicomponent glycoconjugations. The P-3CRs were performed

at 2 mg/mL of protein in PBS at pH 7.4, with a mass ratio of CPs Pn14/CPs MenC/BSA of 2:2:1 and stirring at room temperature for 48 h. These reaction conditions, i.e., CPs/protein ratio, concentration and time, are the typical ones of other glycoconjugation procedures used for vaccine production. All the conjugates were purified by diafiltration with 100 kDa cellulose membranes to remove the excess of unconjugated biomolecules. The multivalent glyco-conjugates BSA-Pn14-MenC **23, 24, 25** and **26** were obtained with a good efficiency considering the substantial amount of recovered glycoconjugate and the unequivocal incorporation of the two CPs into the four glyco-conjugates, as proven by antigenicity analysis (Scheme 4D).

SDS-PAGE with Coomassie and Fuchsin staining corroborated the formation of protein-polysaccharide conjugates of higher molecular weight than the starting isocyanoproteins (Fig. S20 in the SI). Since glucoconjugates **23** and **24** arise from site-selectively tagged isocyano-BSAs, the two poly-saccharide antigens are incorporated at a single residue. To our knowledge, this is a first-in-class type of multivalent protein glycoconjugate, as two polysaccharide antigens are linked to the carrier protein at a single position. In contrast, glycoconjugates **25** and **26** have multimeric structures as they derive from the polyfunctionalized isocyano-BSAs. It is worth mentioning that regardless of the use of site-selectively or polyfunctionalized isocyanoproteins, the resulting glycoconjugates could occur as cross-linked lattice matrices owing to the polyfunctional nature of the two CPs. The polydisperse nature of such a type of protein-polysaccharide conjugate is evident in the bands appearing in the upper part of the electrophoresis gels (Fig. S22). Interestingly, glycoconjugations using polyfunctionalized isocyano-BSAs **12a** and **14** resulted in the consumption of most of the carrier protein, while a certain amount of unconjugated protein remained in the reactions employing the isocyano-BSAs **2b** and **5** site-selectively tagged at the N-terminus and Cys34, respectively. This can be explained by the fact that the first ones have a higher number of isocyanide groups, and therefore better chances to be consumed in the P-3CR than those bearing a single isocyanide handle. Nevertheless, SE-FPLC analysis proved that in all cases, the percentage of free protein in the crude reactions was less than 20%, which is the maximum value recom-mended by the WHO[72] for the production of pneumococcal conjugate vac-cines (Fig. S21 in the SI). Further purification by SE-FPLC allowed reducing the amount of free protein to 1–3% for glycoconjugates **23** and **24**, as exemplified in the SE-FPLC traces shown in Fig. S23. The mass ratio between the three components and the total carbohydrate/protein ratio were deter-mined with the use of colorimetric methods, and were in all cases within the range from 1:3 to 3:1, which is the recommended one by the WHO[69] (Table S3). Given the operational simplicity and success of this bioconjuga-tion protocol, we truly believe that it is scalable and suitable for the devel-opment and production of glycoconjugate vaccines.

The ultimate proof of the success of the Passerini bioconjugation was achieved by evaluating the antigenicity of the conjugates. This was done by a Dot blot assay employing reference antibodies against the BSA, *S. pneumo-niae* and *N. meningitidis*. As shown in Scheme 4D, the recognition of the purified multivalent glycoconjugates **23, 24, 25** and **26** was compared with that of the fragmented/oxidized CPs14, CPsC, and the isocyano-BSAs, as well as with the natural biomolecules as positive controls. Notably, all glyco-conjugates were recognized by the specific antibodies against the protein and each bacterial antigen, confirming that the antigenic determinants of both bacterial CPs were preserved in the multicomponent conjugation. Overall, this immunoassay demonstrated not only the effective multivalent display of the three biomolecular antigens in the unimolecular constructs, but also the actual potential of this strategy in the field of combined conjugate vaccines. The preparation and immunological evaluation in animal models of multi-valent glycoconjugates incorporating other carrier proteins (e.g., CRM197 and tetanus toxoid) and both meningococcal and pneumococcal CPs is beyond the objective of this work, but is certainly projected as part of our ongoing efforts to develop antimicrobial conjugate vaccines.

## Conclusions
We have developed a new multicomponent bioconjugation strategy that employs site-selectively and polyfunctionalized isocyanoproteins for the

one-pot assembly of multivalent (glyco)conjugates using the Passerini reaction. Methods for the single-residue and multimeric derivatization of proteins with isocyanide handles were implemented, resulting in novel isocyanide-tagged proteins functionalized selectively at the N-terminus or a Cys residue. The Passerini bioconjugation was initially tested in protein lipidation and labeling, using palmitic acid and carboxy-fluorescein as the carboxylic acid component and either formaldehyde or acetone as the oxo-component. These experiments proved that acetone is well accepted, but formaldehyde is not a suitable reactant, as it leads to some protein cross-linking. In contrast, the use of other aliphatic alde-hydes led to the efficient incorporation of pharmacologically relevant moieties such as a lipid and a PEG chain into multivalent conjugates, along with carbohydrates reacting with the carboxylic acid group. Elec-trophoresis and mass spectrometry unequivocally confirmed that the conjugation of the oxo and acid components only takes place if the protein is functionalized with the isocyanide handle. The success in the multicomponent conjugation of sialic acid—together with various aldehydes—led us to employ the P-3CR in the assembly of glycoconju-gates that incorporate relevant biomolecular antigens such as the gang-lioside GM3 and the meningococcal C polysaccharide, both containing sialic acid as a key building block. As a final demonstration of the potential of this approach, we produced site-selective and multimeric multivalent glycoconjugates incorporating two bacterial CPs antigens, CPs MenC, providing the carboxylic acid and the periodate-oxidized pneumococcal serotype 14 CPs, providing the aldehyde component. Antigenicity evaluation proved that the three biomolecular components integrated in the multivalent constructs are properly recognized by reference antibodies, confirming the effective display of their epitopes/ antigenic determinants upon the conjugation. The fact that the P-3CR produces a depsipeptide-like linker, i.e., an α-acyloxy-carboxamide moiety, that is not expected to be immunogenic per se, is also positive for potential applications in vaccinology. In summary, the Passerini bio-conjugation is an efficient and versatile approach for the conjugation of two different biomolecules, polymers or labels to a protein at a residue that was previously functionalized with an isocyanide handle. The straightforward construction of unimolecular multivalent conjugates is indeed a powerful tool towards new applications in preventive vaccines and therapeutics.

## Methods
### General procedure for the *N*-terminal functionalization of pro-teins with isocyanide linkers
Isocyanides **1a,b** (50 equiv for BSA, 35 equiv for RNAse A, and 16 equiv for ubiquitin) dissolved in DMF (40 mg mL⁻¹) were added dropwise to the proteins dissolved in 0.1 M carbonate buffer (pH 10.4, $CO_3^{2-}/HCO_3^-$) to reach a final concentration of 10 mg mL⁻¹ and no more than 10% of DMF in the final volume. The reaction mixtures were stirred overnight (16 h) at room temperature. The isocyanide-tagged BSAs **2a,b** were purified by ultrafiltration in Amicon® with a 10 kDa regenerated cellulose membrane using PBS pH 7.4. The isocyanide-tagged RNAse A and ubiquitin were purified by ultrafiltration in Centricon® with 3 kDa regenerated cellulose membrane using PBS pH 7.4. MALDI-TOF characterization is shown in Fig. S18 and SDS-PAGE analysis is shown in Scheme 1.

### Procedure for the functionalization of BSA with tosylate alkyl isocyanide linkers
Isocyanide **11a** or **11b** (50 equiv) dissolved in DMF (40 mg mL⁻¹) was added dropwise to the protein dissolved in 0.1 M carbonate buffer (pH 10.4, $CO_3^{2-}/HCO^{3-}$) to reach a final concentration of 10 mg mL⁻¹ and no more than 10% of DMF in the final volume. The reaction mixture was stirred overnight (16 h) at room temperature. The resulting isocyanoprotein was purified by ultrafiltration in Amicon® with 10 kDa regenerated cellulose membrane using PBS. Isocyanoproteins **12a** and **12b** were characterized by ESI-TOF spectrometry to determine the level of functionalization (see Scheme 2 and Table S1). SDS-PAGE analysis is shown in Fig. S21.

### Procedure for the functionalization of BSA with isocyanide linker 13 via peptide coupling

BSA was dissolved in 0.01 M PBS (pH 7.4) at a concentration of 10 mg mL$^{-1}$, and isocyanide **13** (50 equiv) dissolved in the minimum volume of DMF (40 mg mL$^{-1}$) was added to the protein solution. The reaction mixture was cooled in an ice bath and treated with HATU (50 equiv), then allowed to reach room temperature and stirred overnight (16 h). The resulting isocyanoprotein **14** was purified by ultrafiltration in Amicon® with 10 kDa regenerated cellulose membrane using PBS. Isocyanoprotein **14** was characterized by ESI-TOF spectrometry to determine the level of functionalization (see Scheme 2).

### Synthetic procedure for the functionalization of BSA with maleimido-isocyanide linker

Isocyanide **4** (50 equiv) dissolved in DMF (40 mg mL$^{-1}$) was added dropwise to a solution of BSA in 0.01 M PBS buffer (pH 7.4, with 1 mM EDTA) to reach a final concentration of 10 mg mL$^{-1}$ and no more than 10% of DMF in the final volume. The reaction mixture was stirred overnight (16 h) at 4 °C under a flow of N$_2$ (g). The isocyano-BSA **5** was purified by ultrafiltration in Amicon® with 10 kDa regenerated cellulose membrane using PBS. SDS-PAGE analysis is shown in Scheme 1 and Fig. S21 and MALDI-TOF characterization in Fig. S19.

### General procedure for the conjugation of small molecules to isocyanoproteins *via* the P-3CR

5(6)-Carboxyfluorescein (50 equiv) dissolved in the minimum volume of DMF, and either formaldehyde or acetone (50 equiv) was added to the isocyanoprotein dissolved in 0.01 M PBS (pH 7.4) to reach a final concentration of 2–3 mg mL$^{-1}$ and no more than 5% of DMF in the final volume. The reaction mixture was stirred, protected from light, for 16 h at room temperature. The labeled protein was purified and protected from light by ultrafiltration in Amicon® with 10 kDa cellulose membrane (BSA derivatives) and in Centricon® with 3 kDa cellulose membrane (RNAse A and ubiquitin derivatives) using PBS pH 7.4 until the filtrate was transparent. The negative controls BSA(C-), RNAse A(C-) and ubiquitin (C-) were prepared by mixing and incubating the non-functionalized proteins with carboxyfluorescein under the same reaction and purification conditions explained above. For the reaction with palmitic acid, only 20 equiv of the fatty acid was employed due to solubility issues. MALDI-TOF characterization is shown in Fig. S20, and SDS-PAGE analysis is shown in Scheme 1.

### General procedure for the simultaneous glycosylation and lipidation/PEGylation via P-3CR

A solution of sialic acid (50 equiv) and either lauryl aldehyde or the 2 kDa PEG aldehyde (50 equiv) in the minimum volume of 0.01 M PBS (pH 7.4) was added dropwise to the isocyano-BSA protein dissolved in PBS, to reach a final concentration of 3 mg mL$^{-1}$. The reaction mixture was stirred for 24 h at room temperature. The protein conjugates (**17, 18, 19** and **20**) were purified by ultrafiltration in Amicon® with 10 kDa regenerated cellulose membrane using PBS pH 7.4. When GM3 was used as a carboxylic acid component, the glycolipid was dissolved in 0.01 M PBS (pH 7.4) with octyl-β-glucoside (0.025 M) as detergent, and the final protein concentration was 1.5 mg mL$^{-1}$. The resulting conjugates **21** and **22** were purified by dialysis with a 12 kDa membrane at 4 °C using detergent-free buffer (PBS 0.01 M, pH 7.4), and then by ultrafiltration in Centricon® with a 10 kDa regenerated cellulose membrane using PBS pH 7.4. MALDI-TOF characterization is shown in Scheme 3 and SDS-PAGE analysis is shown in Fig. S21.

### Procedure for the fragmentation and periodate oxidation of the CPs of *Streptococcus pneumoniae* serotype 14 (CPs Pn14)

An aqueous solution of trifluoracetic acid (200 mM) was added to another solution of natural CPs14 (TSK 5000, Kd = 0.184) (2 mg mL$^{-1}$), and the reaction mixture was stirred at 70° C for 30 min. The resulting fragmented polysaccharide was diafiltrated through a series of regenerated cellulose membranes (10–30 kDa) with 10 water volumes each time, and 4 bar nitrogen pressure to afford CPs14 (10–30 kDa, TSK 5000 Kd = 0.572, 88%). The fragmented CPs14 (5 mg mL$^{-1}$) was mixed with NaIO$_4$ (1 mM), and the reaction mixture was protected from light and stirred at 37 °C for 3 h. An excess of ethylene glycol was added to quench the reaction, and the fragmented oxidized CPs14 was purified by diafiltration through a regenerated cellulose membrane (10 kDa) with 10 water volumes and 4 bar nitrogen pressure to afford oxCPs14 containing 1 carbonyl group every 12 repetitive units (RU) (TSK 5000 Kd = 0.388, 65%). This CP was used as an oxocomponent in the Passerini bioconjugation.

### Procedure for the fragmentation of the CPs of *Neisseria meningitidis* serogroup C (CPs MenC)

An aqueous solution of trifluoracetic acid (5 mM) was added to another solution of natural CPsC (2 mg mL$^{-1}$) (TSK 5000, Kd = 0.184), and the reaction mixture was stirred at 60 °C for 2 h and 30 min. The resulting fragmented polysaccharide was diafiltrated through a series of regenerated cellulose membranes (10–30 kDa) with 10 water volumes each time and 4 bar nitrogen pressure to afford CPs MenC (10–30 kDa, TSK 5000 Kd = 0.513, 63%).

### General procedure for the simultaneous conjugation of two bacterial CPs to a carrier protein via P-3CR

The pneumococcal and meningococcal CPs, reacting as carbonyl and carboxylic acid components, respectively, were mixed in a 1:1 (w/w) ratio and dissolved in the minimum volume of buffer PBS (0.01 M, pH 7.4). Then, the isocyanoprotein dissolved in buffer PBS was added dropwise in a 2:1 (w/w) ratio related to the mass of the carbonyl-containing CPs to reach a final concentration of 2 mg mL$^{-1}$. The reaction mixture was stirred for 72 h at room temperature. The product was purified by ultrafiltration with 100 kDa regenerated cellulose membrane using PBS pH 7.4 and, if necessary, by SE-FPLC.

### Antigenicity assessment of the isocyano-BSAs and glycoconjugates

The native BSA protein, the isocyano-BSAs, the CPs and their glycoconjugates (5 µg of CPs or protein) were applied on a nitrocellulose membrane and incubated in a 5% skim milk solution in TBS (20 mM Tris, 500 mM NaCl, pH 7.5) at 37 °C for 30 min. The membrane was washed three times for 5 minutes with a washing solution containing TBS and Tween 20 (0.05%). Next, the membrane was incubated with the primary antibodies: anti-BSA rabbit polyclonal IgG antibody (1:1000, Invitrogen Ref A11133, Lot 2206803), anti-*S. pneumoniae* rabbit polyclonal IgG antibody (1:1000, Bio-Rad Ref 0300-0218, Lot 155340, reactive for serotype 14) and anti-*N. meningitidis* rabbit polyclonal IgG antibody (1:2000, Bio-Rad Ref 6600-5906, Lot 158003, reactive for serogroup C) in a 1% skim milk solution in TBS at 37 °C for 60 min. After washing the membrane three times for 5 min with the washing solution, it was incubated with an anti-rabbit IgG antibody conjugated to HRP (1:10 000, Santa Cruz SC2357) in TBS at 37 °C for 60 min. Finally, the membrane was washed three times for 10 min with the washing solution and revealed using Pierce Chemiluminescent Kit and OPTIMAX 2010 X-Ray Film Processor.

### Reporting summary

Further information on research design is available in the Nature Portfolio Reporting Summary linked to this article.

### Data availability

The authors declare that all the data supporting the findings of this study are available within the article and Supplementary Information files, and are also available from the corresponding authors upon reasonable request.

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

## Acknowledgements

A.R.H. and Y.M. are grateful to DAAD, Germany, for doctoral fellowships. We also acknowledge financial support from DAAD through the GLACIER project (57592717).

## Author contributions

A.R.H. and Y.M.: conceptualization, methodology, investigation, data curation and writing. A.K., M.N., A.F., M.S., R.G, and L.R.: investigation, data curation, reviewing and editing. D.S.-M., and D.G.-R.: methodology, validation, reviewing and editing. B.W. and D.G.R: conceptualization, methodology, supervision, funding, writing, reviewing and editing. All authors have given approval to the final version of the manuscript.

## Funding

## Competing interests

The authors declare no competing interests
