## [Transparent Peer Review file · Communications Chemistry]

Creating Unimolecular Multivalent Diversity in Protein Conjugates via the Passerini Multicomponent Bioconjugation with Isocyanoproteins

Corresponding Author: Professor Daniel Rivera

Version 0:

Reviewer comments:

Reviewer #1

(Remarks to the Author)

The authors of this article have developed an efficient method for the functionalization of BSA in a selective or non-selective manner.

In a first step, the protein was modified to introduce isonitrile residues. These isonitrile-functionalized proteins were then used as substrates in Passerini-type multicomponent reactions with various acid- and aldehyde-containing molecules of biological relevance.

Although the Passerini reaction has previously been used for protein functionalization, the methodology proposed in this work is innovative, as it does not rely on the naturally occurring functional groups of the protein. This allows for greater control over the bioconjugation process.

I found the article well written and clearly structured. The topic is highly interesting, and the results presented suggest a promising potential for applying this strategy in the field of synthetic vaccines. The cited literature appears appropriate, and the experimental section is carefully detailed. For these reasons, I recommend acceptance of the manuscript, with only a few minor suggestions:

In the supplementary material, I suggest clarifying whether the purified and isolated reaction intermediates are known compounds or not. If they are not, their characterization should be provided; if they are, the corresponding literature references should be clearly indicated.

The NMR spectra copies should be standardized by including the signal frequencies and by expanding the spectral window (around 11 to -0.5 ppm for ^1H , and 220 to -10 ppm for ^{13}C).

It could be helpful for some readers to know the formulation of the Ni(II)-based stain used to detect isonitriles on TLC. Could you consider adding it to the general methods section?

Reviewer #2

(Remarks to the Author)

The study presents a novel, one-pot multicomponent bioconjugation strategy using the Passerini three-component reaction (P-3CR) with isocyanide-tagged proteins to create unimolecular multivalent protein conjugates. The team has reported a similar approach using the so called Ugi four-component reaction to assemble multicomponent unimolecular conjugates (Chem. Sci. 2018, 9, 2581-2588).

This work aims at streamlining the process of attaching two different biomolecules to proteins, typically requiring multi-step processes with challenging purification steps, by using a one-pot protocol.

Proteins like bovine serum albumin (BSA), RNase A, and ubiquitin were modified either site-selectively (at the N-terminus or a free cysteine residue) or in a polyfunctional manner by incorporating isocyanide handles. These handles then enabled the simultaneous conjugation of two diverse components—such as fluorescent dyes, lipid tails, polyethylene glycol (PEG), and various carbohydrate antigens, including sialic acid, ganglioside GM3, and bacterial capsular polysaccharides—via the P-3CR under aqueous conditions. Optimized conditions were used to finally assemble a biomolecule containing two different polysaccharide (pneumococcal type 14 and meningococcal type C). The conjugated proteins were characterized by electrophoresis and mass spectrometry, while antigenicity was validated via dot blot assays, confirming the effective display of biomolecular epitopes.

This procedure requiring mild conditions seems expands the toolbox for developing next-generation conjugate vaccines and protein therapeutics by enabling the rapid assembly of structurally diverse, multivalent conjugates in a single synthetic step. The article is well written and the work well executed. There are some points whoever that can be improved to further support the conclusions.

- Isocyanates react rapidly with thiol, amine, or hydroxyl groups although the reactions is expected to be faster for thiol. Incorporation of a single moiety in construct 10 in Cys34 rather the N-terminal group should be supported by MS data on peptides from digested proteins.

- The bivalent conjugates in reality could derive from reaction of two COOH form the same polysaccharide with the isocyanate. To rule out this authors run reactions with sialic acid and defined lipid or PEG molecules, showing in Scheme 3 zoom on the expected products. Can authors add as supplemental information the full spectra to exclude that minimal amounts of products with two sialic or PEG or lipid adducts are formed?

- Can authors comment on how many polysaccharides bear functional groups that are amenable for this reaction?

- Can authors comment on the excess of polysaccharide used in reaction vs a classic approach as the direct reaction of the generated aldehyde with the lysine residue? Can yields be estimated? Is the approach scalable in the author's opinion?

Version 1:

Reviewer comments:

Reviewer #1

(Remarks to the Author)

Dear authors,

I have reviewed the revised manuscript, which had already undergone a previous review. I confirmed that the authors have implemented the recommended changes, and I consider the article ready for publication.

Best Regards

Reviewer #2

(Remarks to the Author)

The authors have well addressed the comments from the reviewers and the manuscript is now suited for publication.

We are glad that the referees see the novelty of these results and their potential impact in the field of vaccines and therapeutics. We are pleased to confirm that we have been able to make most of changes and corrections suggested by them. In some cases, the points raised were due to misinterpretation of the manuscript or confusion about the reactivity of the functional group used in the paper. In these cases, we preferred to keep our original text. We appreciate their time and effort around this publication.

Response to Reviewer 1:

Comment. In the supplementary material, I suggest clarifying whether the purified and isolated reaction intermediates are known compounds or not. If they are not, their characterization should be provided; if they are, the corresponding literature references should be clearly indicated.

Response: We agree with making changes in the Supplementary Information to make clear which intermediates and final compounds were characterized and which not. Please notice that some intermediates were not characterized by NMR because they are produced by simple reactions like methyl ester formation, saponification or formylation. These were simple and clean reactions that yielded a single product, so they were followed only by TLC and the intermediates were not characterized by NMR. However, we confirm that the final isocyanide compounds were indeed characterized by NMR and HR-MS, as these were used for the protein modification. In the synthetic procedures, we now specify whether the intermediate was purified by CC and characterized or used without further purification and characterization. In the revised version, the NMR and HR-MS data for final new compounds is provided.

Comment. The NMR spectra copies should be standardized by including the signal frequencies and by expanding the spectral window (around 11 to -0.5 ppm for ^1H , and 220 to -10 ppm for ^{13}C).

Response: We agree. We are now showing the signal frequencies in all NMR spectra. We also confirm that the full spectra are provided showing all signals present. We have done this in the standard way for a supplementary information of a chemical paper.

Comment: It could be helpful for some readers to know the formulation of the Ni(II)-based stain used to detect isonitriles on TLC. Could you consider adding it to the general methods section?

Response: We agree. We have also added a Methods section in the main manuscript and have added in the supplementary methods the protocol for staining isocyanides on TLC. Briefly.

Isocyanide identification using Ni(II)-staining

A 2% solution of $\text{NiCl}_2 \cdot 6\text{H}_2\text{O}$ in ethanol was used for the qualitative identification of isocyanides on TLC, which is based on the reaction of Ni(II) ions with the isocyanide group to form colored coordination complexes. The TLC plate is immersed in the Ni(II) solution and then warmed gently with a heating gun until the appearance of a reddish-brown or violet-colored spot, which indicates the presence of an isocyanide group in the molecule.

Response to Reviewer 1:

Comment. Isocyanates react rapidly with thiol, amine, or hydroxyl groups although the reactions is expected to be faster for thiol. Incorporation of a single moiety in construct 10 in Cys34 rather the N-terminal group should be supported by MS data on peptides from digested proteins.

Response. Unfortunately, we cannot satisfy this request, because our paper does not deal with isocyanates at all, but with isocyanides, which is a different functional group that does not react with the above-mentioned functional groups. We however understand the doubt regarding the possible reaction of BSA by the N-terminus. In this case, the chemoselective derivatization of BSA with maleimide reagents at Cys34 is a state-of-the-art procedure in protein chemistry. It is known that under standard thiol-maleimide chemistry at neutral pH (i.e., PBS at pH 7.4, 4 °C) only the sulfhydryl group of Cys34 reacts with the maleimide. In page 11, we now cite three different references (62, 63 and 64) that confirm the selectivity of the maleimide addition to the Cys thiols and not the protein amino groups.

Comment. The bivalent conjugates in reality could derive from reaction of two COOH from the same polysaccharide with the isocyanate. To rule out this authors run reactions with sialic acid and defined lipid or PEG molecules, showing in Scheme 3 zoom on the expected products. Can authors add as supplemental information the full spectra to exclude that minimal amounts of products with two sialic or PEG or lipid adducts are formed?

Response. Once again, the possibility proposed by the referee is impossible because we don't have isocyanates in our proteins. In the conditions reported by us and many others, the isocyanide group can only react with a carbonyl compound and carboxylic acid, one of each, not with two carboxylic acids. This misinterpretation comes from a referee's confusion between isocyanates and isocyanides, which is common in the literature.

Comment. Can authors comment on how many polysaccharides bear functional groups that are amenable for this reaction?

Response. This is already explained in the manuscript, please see page 19, where we say: ‘As shown in scheme 4A, we chose the capsular polysaccharide (CPs) of *S. pneumoniae* serotype 14 (Pn14) – which is present in all pneumococcal conjugate vaccines – to be employed as oxo component. To this end, CPs Pn14 was fragmented to a molecular size of 10-30 kDa and further subjected to periodate oxidation, resulting in approximately one carbonyl group every 10 repetitive units’.

In the same paragraph, we refer to the MenC polysaccharide, which already contains the required carboxylic acids. ‘In parallel, the CPs from the serogroup C of *N. meningitidis* CPs MenC was fragmented under mild acidic conditions, resulting in a 10-30 kDa polysaccharide antigen already containing the required carboxylic acids (Scheme 4).’

Comment. Can authors comment on the excess of polysaccharide used in reaction vs a classic approach as the direct reaction of the generated aldehyde with the lysine residue? Can yields be estimated? Is the approach scalable in the author’s opinion?

Response. This is an interesting point and we agree with adding some comments about it. See in page 19, we now added the following sentences: ‘These reaction conditions, i.e., CPs/protein ratio, concentration and time, are the typical ones of other glycoconjugation procedures used for vaccine production’.

Regarding the efficiency, in page 21 we declare the following: ‘Nevertheless, SE-FPLC analysis proved that in all cases, the percentage of free protein in the crude reactions was less than 20%, which is maximum value recommended by the WHO for the production of pneumococcal conjugate vaccines (Fig. S21 in the SI). Further purification by SE-FPLC allowed reducing the amount of free protein to 1-3% for glycoconjugates 23 and 24, as exemplified in the SE-FPLC traces shown in scheme 4E.’

We have also added a final comment giving our opinion on the possibility to use this chemistry for vaccine production. Please see first paragraph in page 22, we comment the following: ‘Given the operational simplicity and success of this bioconjugation protocol, we truly believe that it is a scalable and suitable for the development and production of glycoconjugate vaccines’.